# Propylene Glycol Potentiates the Inhibitory Action of CTZ Paste on Antibiotic-Resistant *Enterococcus faecalis* Isolated from the Root Canal: An In Vitro Study

**DOI:** 10.3390/microorganisms11092208

**Published:** 2023-08-31

**Authors:** Jesús Yareli Rayos-Verdugo, Fernando Rivera-Chaparro, Gloria Yolanda Castro-Salazar, Maricela Ramírez-Álvarez, José Geovanni Romero-Quintana, Juan Pablo Loyola-Rodríguez, Norma Verónica Zavala-Alonso, Mariana Avendaño-Félix, Jesús Eduardo Soto-Sainz, Erika de Lourdes Silva-Benítez

**Affiliations:** 1Especialidad de Odontopediatría, Facultad de Odontología, Universidad Autónoma de Sinaloa, Culiacán 80040, Mexico; yarelirayos@hotmail.es; 2Laboratorio de Microbiología, Facultad de Ciencias Químico-Biológicas, Universidad Autónoma de Sinaloa, Culiacán 80040, Mexico; ferrvra@gmail.com (F.R.-C.); geovanniromero@uas.edu.mx (J.G.R.-Q.); 3Especialidad de Endodoncia, Facultad de Odontología, Universidad Autónoma de Sinaloa, Culiacán 80040, Mexico; yolandacastro@uas.edu.mx (G.Y.C.-S.); dra.maricela_odontologia@uas.edu.mx (M.R.-Á.); est.marianaavendano@uas.edu.mx (M.A.-F.); eduardosotosainz@uas.edu.mx (J.E.S.-S.); 4Maestría en Rehabilitación Oral Avanzada, Facultad de Odontología, Universidad Autónoma de Sinaloa, Culiacán 80040, Mexico; juanpablo.loyola8@gmail.com; 5Maestría en Odontología Integral del Niño y el Adolescente, Facultad de Odontología, Universidad Autónoma de Sinaloa, Culiacán 80040, Mexico; 6Maestría en Ciencias Odontológicas, Facultad de Estomatología, Universidad Autónoma de San Luis Potosí, San Luis Potosí 78210, Mexico; nveroza@fest.uaslp.mx

**Keywords:** CTZ paste, grapefruit-seed extract, propylene glycol, antibiotic resistance, *E. faecalis*

## Abstract

This study aimed to evaluate if the change of vehicle for CTZ (Chloramphenicol, Tetracycline, zinc oxide, and Eugenol) paste improves the inhibition of *Enterococcus faecalis* in vitro. The vehicles evaluated alone and mixed with CTZ were Eugenol, propylene glycol (PG), super-oxidized solution (SOS), grapefruit-seed extract (GSE), and 0.9% saline solution as a negative control. A clinical isolate of *E. faecalis* was morphologically and biochemically characterized, and its antimicrobial susceptibility was tested using 20 antimicrobial agents. Once characterized, the clinical isolate was cultivated to perform the Kirby–Bauer disc diffusion method with paper discs embedded with the different vehicles mixed or used alone, and incubated at 37 °C for 24 h. Data were analyzed using one-way ANOVA, and the means were compared using Tukey test with a significance level of *p* < 0.05. For vehicles used alone, GSE presented the greatest inhibition showing a statistically significant difference with the rest of the vehicles. When vehicles were mixed with the CTZ paste, PG showed a greater inhibition with a statistically significant difference from the rest of the vehicles. In conclusion, the vehicle used to mix the CTZ paste plays an important role in the inhibition of *E. faecalis* in vitro; therefore, we consider that this can be an important factor to achieve success in the use of this technique.

## 1. Introduction

Dental caries is an important disease in children, being the most prevalent disease in oral cavity at this stage of life [1]. Nowadays, it has been reported in some countries with a prevalence greater than 90% in six year olds, which is the reason authors have considered this a public health crisis [2].

This disease destroys dental hard tissue, affecting pulp vitality, with a high risk of the development of pulp and periapical lesions [3]. To eradicate this polymicrobial infection with aerobic and anaerobic bacteria [4,5], the non-instrumentation endodontic treatment (NIET) has been proposed. This therapeutic approach proposes the use of an antibacterial drug mixture placed over the pulp floor [6]. The aim is to prevent an over-instrumentation of root canals and the irritation of periapical tissue, decreasing chair time to only one visit [7], since endodontic treatment of primary teeth could be a challenge to the presence of root resorption and the successor tooth [6].

Antibiotic pastes have been proposed in endodontic therapy due to their antimicrobial capacity and low cost [8]. If infected primary teeth are not treated effectively, premature tooth extraction could be necessary, affecting masticatory function, maxillofacial and systemic growth [9], and increasing the risk of malocclusions [10]. Several antibiotics have been used in the preparation of these pastes, such as Ciprofloxacin, Metronidazole, Minocycline, Clindamycin [11], Ornidazole [12], or Rifampicin [13]. Interestingly, a mixture of Chloramphenicol 500 mg, Tetracycline 500 mg, zinc oxide 1000 mg, and one drop of Eugenol (CTZ) in a ratio of 1:1:2 was introduced by Cappiello in 1964, showing promising results [14]. The success of this paste is due to its composition since Chloramphenicol is a broad-spectrum antibiotic [15] with high solubility and bacteriostatic effect against Gram-positive anaerobic and Gram-negative aerobic bacteria [16]. The Tetracycline is a broad-spectrum bacteriostatic agent that inhibits bacterial protein synthesis [17]. Also, the antimicrobial capacity of the zinc oxide is supposedly affected by the production of hydrogen peroxide [18]; however, some authors have reported that the zinc oxide alone has no inhibitory effect, and it is the Eugenol which has the antimicrobial action [19,20].

In the clinic, several other types of pastes are used to disinfect the root canal system and create a favorable environment for endodontic regeneration. Triple Antibiotic Paste (TAP), composed of Ciprofloxacin, Metronidazole, and Minocycline [21,22], is a well-studied example. Studies have shown that the vehicles used with TAP can influence cytotoxicity [23,24,25]. For instance, the use of the combination of Polyethylene Glycol and Propylene Glycol (PG) mixed as vehicle exhibit less cytotoxicity of TAP than when water is used [26]. In addition, the change of vehicle can interfere in the antimicrobial capabilities of the TAP since some of them are able to improve this property against some microorganisms [27].

Currently, it has been reported that antibiotic resistance decreases the antibacterial properties of endodontic filling pastes for root canal treatment [13], which highlights the importance of looking for vehicles that enhance the antimicrobial effects of these pastes. In the CTZ paste, the conventional vehicle recommended by Cappiello is the Eugenol [14], which has even been reported to cause periapical irritation [28]. However, other type of substance can be tested to improve the antimicrobial capacity of CTZ paste, such as PG, a dihydric alcohol [29] with bactericidal activity [30], and which, as a vehicle, has demonstrated to improve the diffusing property of drugs [31]. Interestingly, a super-oxidized solution (SOS), which is an electrochemically processed solution made from water and NaCl, presents antimicrobial capacity since it has been proposed to clean root canal walls and is capable to eradicate *Enterococcus faecalis* (*E. faecalis*) [32,33,34].

Nowadays, there is a tendency to evaluate natural products with antimicrobial activity against planktonic and biofilm microorganisms for therapeutic dentistry [35]. Since we face an antibiotic resistance crisis [36] in modern times, it is necessary to consider options that improve, in this case, the antibiotic activity of CTZ compounds, allowing for a possible synergic effect. Some of their advantages are biocompatibility, antioxidant, anti-inflammatory [37], and healing properties [38], as well as being side-effect free [35]. Therefore, the natural therapeutic agent propolis was previously used as a vehicle of TAP, resulting in an improvement of the antimicrobial and regenerative properties [38]. The reported action mechanism of natural products is cell wall or membrane destruction, energy synthesis, biofilm formation inhibition, and a production stimulation of reactive oxygen species [39].

Interestingly, the grapefruit-seed extract (GSE) is a natural product obtained by grinding the seeds, pulp, and white membranes from *Citrus paradisi* and mixing them with glycerin [40]. This extract has shown a powerful antimicrobial activity [41] mainly attributed to the presence of polyphenolic compounds, flavonoids, citric and ascorbic acid, tocopherol, and limonoid [42].

Several microorganisms, such as *E. faecalis*, are present in endodontic infections. This one has been widely studied due to its resistance to conventional endodontic treatment [43], being one of the most important bacteria on the root canal [44]. *E. faecalis* is a Gram-positive facultative anaerobe cocci bacterium that inhabits the human oral cavity, gastrointestinal tract, and vagina [45]. This bacterium has the capacity to invade dentinal tubules due to the presence of a collagen-binding protein (Ace) [46], resist nutritional deprivation [47], and delay the penetration of antimicrobial agents due to the presence of its enterococcal surface protein (*Esp*), producing a biofilm of polystyrene [48], as well as surviving temperatures of 10° to 60 °C, avoiding the action of lymphocytes [49], with the ability to grow in high pH ambience [50]. It even has the capacity to form a biofilm in the presence of Ca(OH)_2_ solutions [51], mostly for its ability to attach to abiotic surfaces [52] and other bacteria, serum, collagen, and dentin [53]. Hence, the virulence factors of *E. faecalis*, as adhesion and colonization, resistance to host defense, inhibition on other bacteria, tissue damage, and induction of inflammation summarized by Kayaoglu et al. [53], emphasize the necessity to find strategies to manage the elimination of this microorganism.

Until now, no studies have been found in the literature evaluating if the change of vehicle in this paste can modify the drug delivery and potentiate its action. Thus, the aim of this study was to evaluate if these products inhibit *E. faecalis* by themselves or have the capacity to potentiate the antimicrobial effect of CTZ paste.

## 2. Materials and Methods

### 2.1. Materials

The evaluated vehicles were PG (Fagalab^®^, Mocorito, Sinaloa, México), SOS (Esteripharma^®^, Ciudad de México, Ciudad de México, México), and grapefruit-seed extract (Distribuidora Hidalgo, Naucalpan, Estado de México, México). Eugenol (ViardenLab^®^, Mission, TX, United States of America) and 0.9% saline solution (S.S.) (PISA, Culiacán, Sinaloa, México) were used as positive and negative controls, respectively. The CTZ (Farmacia Galenico, Zacatecas, Zacatecas, México) was commercially obtained to avoid the variation in the formulation and the presence of excipients. The bacterium *E. faecalis* was donated by the Research Department of Endodontics of the Stomatology Faculty of the Autonomous University of San Luis Potosí where it was isolated from the root canal of patients with secondary endodontic infections employing sterile paper tips, and subsequently cultivated in thioglycolate tubes that were incubated in an anaerobic chamber [54].

### 2.2. E. faecalis Characterization

*E. faecalis* was characterized using Gram staining (Hycel^®^, Zapopan, Jalisco, México) for the determination of cell morphology and the classification in Gram-positive or Gram-negative bacteria using optical microscopy. Then, biochemical tests with twenty-six substrates and the antibiotic resistance of 20 drugs were determined using the kit Microscan pos combo panel type 33 (Beckman coulter, Pasadena, United States of America) following the manufacturer’s instructions.

### 2.3. Growth Kinetic of E. faecalis

*E. faecalis* was cultivated in Tryptic Soy Broth (TSB) (BD, Franklin Lakes, NJ, United States of America) medium evaluating absorbance at 600 nm using an Optizen pop (KLAB, Yuseong-gu, Daejeon, Republic of Korea) every hour for 12 h to establish growth kinetics. Additionally, serial dilutions were made in Trypto-casein Soy Agar plates (TSA) to evaluate colony forming units (CFUs), evaluating cell viability during the growth kinetics.

### 2.4. Drug Preparations

An amount of 500 mg of CTZ was mixed with the vehicles, using a different amount of each (Table 1), to obtain a paste with a firm and adhesive consistency (Figure 1).

### 2.5. Kirby–Bauer Disc Diffusion Method

The *E. faecalis* was adjusted to 0.5 McFarland standard by adding sterile distilled water, which corresponds to ~1.5 × 10^8^ cells/mL. The bacterial suspension of 100 µL was distributed throughout the plate with Müller Hilton Agar (BD, Franklin Lakes, NJ, United States of America) to place a triplicate of absorbent paper discs of 5 mm in diameter impregnated with each mixture or a triplicate of absorbent paper discs impregnated only with 5 µL of each different vehicle. And, as a control, discs impregnated with S.S. were employed. All plates were incubated 24 h at 37 °C. After incubation, the plates were observed, and the inhibition zone was measured with a digital vernier (Truper^®^. Ciudad de México, México).

### 2.6. Statistical Analysis

For the data analysis, the means and standard deviation of the inhibition zone size were used. One-way ANOVA and Tukey tests were used with a significance level of *p* < 0.05 using GraphPad prism v.8.

## 3. Results

### 3.1. E. faecalis Exhibit Resistance Behavior to Antibiotics

The clinical isolate of *E. faecalis* showed the typical cell morphology of these bacteria as Gram-positive cocci in pairs and short chains on Gram stain (Figure 2A). Also, the biochemical characterization showed that the clinical isolate has a 99.9% of correspondence with *E. faecalis* (Table 2). Furthermore, this characterization was complemented by the resistance to antibiotics analysis (Table 3), observing that this bacterium was sensitive to Ampicillin, Ciprofloxacin, Daptomycin, Gentamicin, Penicillin, and Rifampicin, showing intermediate resistance to Linezolid. Interestingly, the clinical isolate was resistant to Erythromycin, Streptomycin, and Tetracycline, the latter being one of the main components of CTZ paste, highlighting the importance of improving the antimicrobial effect of CTZ paste.

### 3.2. E. faecalis Has a Conventional Pattern of Growth

The results of the bacterial calibration curve using OD showed that *E. faecalis* has a lag phase duration of 3 h. The logarithmic growth starts at hour 3 and continues for 6 more hours when the bacteria enter into the stationary phase (Figure 2B). Viability evaluation was performed, counting the CFU each hour in a period of 13 h, observing that the growth peak for *E. faecalis* starts at hour 4 (Figure 2C), so, at this time, a bacteria aliquot was taken to carry out the antibiogram test with the CTZ paste mixed with different vehicles.

### 3.3. PG Potentiates CTZ Paste Effect on E. faecalis

The results of the inhibition of bacterial growth were obtained by mixing CTZ paste with different vehicles, which, after 24 h, showed that the bacterial growth of the paste was not potentiated when mixed with SOS (32.6 ± 2.0 mm, *p* = 0.2000) and GSE (26.8 ± 2.5 mm, *p* = 0.9891), with no significant difference to positive-control Eugenol (30.8 ± 0.1); however, PG did show a greater inhibition of bacterial growth (36.9 ± 1.0 mm, *p* = 0.0021) compared to the positive control (Figure 3A). No significant difference was found when Eugenol was compared to the negative-control S.S. (26.8 ± 2.5 mm, *p* = 0.1542), meaning that this vehicle does not promote a real antibacterial capacity of the CTZ paste.

### 3.4. GSE Has a Potential Antibacterial Effect Used Alone

To evaluate if the vehicles used enhanced the effect of the CTZ, the Kirby–Bauer disc diffusion method was employed, finding that S.S has a null antibacterial effect. However, PG (13.8 ± 0.8, *p* = 0.9987) has a similar effect to Eugenol on *E. faecalis* (14.2 ± 0.3), and a major inhibition was found using SOS (20.3 ± 1.5, *p* = 0.0023) and GSE (30.9 ± 3.6, *p* < 0.0001), the latter being the vehicle with the highest antibacterial effect (Figure 3B), proposing GSE as a solution that could be used alone as an irrigant.

### 3.5. GSE Extract Does Not Need Antibiotics to Improve Its Bactericidal Capacity

The comparison of the inhibition of the bacterial growth of the vehicle alone and mixed with CTZ evidenced that the antibacterial effect, shown by the S.S. (*p* < 0.0011) mixed with the paste, is due to its components, since used alone did not show any inhibition in the growth of *E. faecalis* (Figure 3C). Nevertheless, the conventional vehicle Eugenol from the CTZ (*p* < 0.0001), as well as SOS (*p* = 0.0011) and PG (*p* < 0.0001) increased its antibacterial potential once mixed with the paste. No significant differences were found when comparing GSE used alone or mixed with the paste (*p* = 0.9762). Therefore, it could be used alone as an irrigant solution since antibiotics are not necessary to improve its bactericidal capacity.

## 4. Discussion

It has been reported that the presence of *E. faecalis* is similar in temporary and permanent teeth [55], so it is important to look for alternatives that allow the clinician to eliminate this microorganism. Currently, the use of a mixture of broad-spectrum antibiotics has been introduced as a pulp therapy treatment called CTZ paste [56], which has exhibited the capacity to eliminate *E. faecalis* with the use of Eugenol as a vehicle in vitro [15,57,58]. However, it has been reported that the change in the vehicle can improve the diffusion and release of this drug [59]; thus, this study aimed to assess if the change of vehicle for CTZ paste improves the inhibition of *E. faecalis* in vitro.

To analyze the antimicrobial susceptibility, we employed a wide range of antibiotics, showing that our clinical isolate *E. faecalis* exhibited resistance to a large number of them, such as Erythromycin, Streptomycin, and Tetracycline; however, some antibiotics tested showed effectiveness against *E. faecalis*, such as Ampicillin, Ciprofloxacin, Daptomycin, Gentamicin, Penicillin, and Rifampicin. The aforementioned result differs from Pazhouhnia et al. [60] since they reported a different pattern of antibiotic resistance in several *E. faecalis*-isolated strains from the root canal of individuals with periodontitis, demonstrating that both isolated strains, theirs and our clinical isolate of *E. faecalis*, are resistant to Tetracycline, one of the main components of CTZ paste. This highlights the variability in pathogenicity that can be found in different clinical isolates.

Also, the bactericidal capacity of Eugenol is due to its ability to produce hydrophobicity, which alters the cell membrane, making it more permeable [20], leading to the extreme loss of molecules and ions and finally to cell death [61]. Despite this, Eugenol shows in our results a medium inhibition alone and mixed with CTZ compared with the rest of the vehicles. Similar results have been observed by De Sales Reis et al., who reported a similar inhibition of *E. faecalis*, employing CTZ paste mixed with Eugenol [58]. Other authors reported a greater potentiating effect of Eugenol as a vehicle on *E. faecalis* [15]; nevertheless, they used a clinical isolate and, as we mentioned above, these may differ in their pathogenicity.

In addition, propylene glycol (1,2-propanediol) has demonstrated to improve the diffusing property of several drugs [31]. Remarkably, it has bactericidal activity against several bacteria, such as *Streptococcus mutans*, *Escherichia coli*, and even *E. faecalis* [30]. In our results, this vehicle used alone showed a Eugenol-like effect against *E. faecalis*. However, our results differ from other studies, such as by Thomas et al., who found that Eugenol presented antibacterial properties against *E. faecalis*; meanwhile, PG did not show the same effect on this bacterium [62]. This behavior has already been reported before in other studies [63], proposing an erratic antibacterial effect of the PG used alone.

On the other hand, in the context of its function as a vehicle, PG mixed with endodontic medicament has shown antimicrobial activity against several bacteria, such as *S. aureus*, *E. coli*, and including *E. faecalis* in vitro [59]. Previously, it had been described that, mechanically, PG as a vehicle potentiates the penetration of the drugs into the dentinal tubules [29]. In this context, Pereira et al. evaluated this vehicle with TAPs against a reference strain of *E. faecalis,* showing intratubular decontamination in vitro [63]. In our study, when this vehicle was mixed with CTZ, it achieved the highest halo of inhibition compared to the rest of the vehicles. This could be related to the hygroscopic nature and viscosity of PG, which allow the sustained release of ions increasing the antibacterial properties of the drugs [64]. Therefore, our results showed that this vehicle allows the optimal release of the CTZ active principles.

In addition, super-oxidized solutions have been shown to be powerful antimicrobial agents and disinfectants through oxidative damage [65] since electrolyzed water contains a mixture of inorganic oxidants, such as hypochlorous acid (HClO), hypochlorous acidic ion (ClO^−^), chlorine (Cl_2_), hydroxide (OH), and ozone (O_3_). However, controversial results have been described about the SOS antimicrobial capacity since this solution exhibits an erratic effect in the elimination of the *E. faecalis* from the root canal compared to NaOCl [33,34]. Our results showed that SOS used alone may have an intermediate antibacterial effect against our clinical isolate of *E. faecalis*; however, its properties as a vehicle are not remarkable compared to the other materials evaluated.

GSE is a natural product obtained from *Citrus paradisi* by grinding its seeds, pulp, and white membranes and mixing them with glycerin [40]. This extract has shown a powerful antimicrobial activity [41] mainly attributed to the presence of polyphenolic compounds, flavonoids, citric and ascorbic acid, tocopherol, and limonoid [42]. This capacity has been evaluated against ATCC reference strains of *E. faecalis,* showing low [41] to high [66] antibacterial effects against this bacterium. In our study with *E. faecalis* clinical isolate, a great inhibition was observed, which could be related to GSE concentration since previous studies were performed at a concentration of 33%; meanwhile, we employed a higher concentration (46%). Although this extract by itself obtained the highest inhibition zone compared with the rest of the vehicles, the result was not the same when mixed with CTZ, showing a similar inhibition to the recommended vehicle Eugenol; however, until now, no studies have reported the evaluation of this extract as a vehicle for CTZ. We observed that it does not improve the antimicrobial activity of the paste; therefore, we consider that it is not the ideal vehicle for the application of these antibiotics.

GSE could be used alone as an irrigation solution since it has demonstrated antimicrobial activity against Gram-negative and -positive bacteria and yeast [41]. Previously, this function of GSE as an irrigant had been evaluated, finding an effective in vitro action of GSE in the inhibition of biofilms formed by *Porphyromonas endodontalis* and *Porphyromonas gingivalis*, showing a promising antibiofilm activity. Also, a cytotoxicity assay with human gingival fibroblast was carried out observing high biocompatibility, absent characteristic in NaOCl, since it can cause a burning sensation in gingiva and periapical tissue necrosis [67]. This finding, together with our results, is very interesting since the GSE could be used in primary and permanent endodontic procedures that include periodontal tissues without side effects. Another putative advantage of this natural product is that the bacterium does not show resistance to this product since it is a new substance, whereby this resistance is a persistent complication in endodontic treatment, and which has become a global issue [13]. Therefore, it is promissory to use it as a single solution against *E. faecalis* and other bacterial strains present in the oral microbiome associated with active dental infections. However, further experiments need to be performed inside the root canal.

Due to the latter, green technology extracts could be an excellent option against the oral microbiome associated with active dental infections. Furthermore, it is an excellent option to use plant or fruit extracts to clean the root canal system from the irrigation phase. Currently, the use of blueberry and wild strawberry extracts [68] and enzymes from papaya, orange, and pineapple peels [69] have demonstrated promising results against *E. faecalis* in vitro, placing them as alternatives to sodium hypochlorite (NaOCl) to avoid severe injuries that might occur with NaOCl accidents or in allergic patients [70].

## 5. Conclusions

The vehicle used to mix the CTZ paste plays an important role in the effect of inhibition against *E. faecalis* in vitro. And although this is not the only microorganism that the clinician faces during pulp therapy, it is one of the most resistant. Therefore, its eradication is extremely important. Our results showed that PG could be the best vehicle for the CTZ paste. Henceforth, it is necessary to evaluate if PG mixed with CTZ maintain a higher inhibition than Eugenol against other microorganisms related to pulpal pathologies. Furthermore, we proposed GSE as an irrigation solution although further experiments need to be performed to confirm this.

## Figures and Tables

**Figure 1 microorganisms-11-02208-f001:**
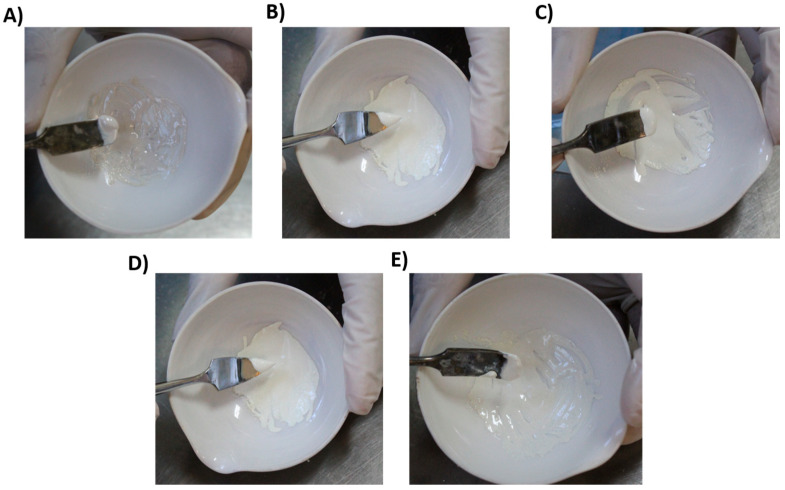
Prepared pastes with different vehicles. (**A**) Eugenol + CTZ +, (**B**) *Saline solution + CTZ*, (**C**) *super-oxidized solution + CTZ*, (**D**) *grapefruit-seed extract +* CTZ, and (**E**) *propylene glycol* + CTZ.

**Figure 2 microorganisms-11-02208-f002:**
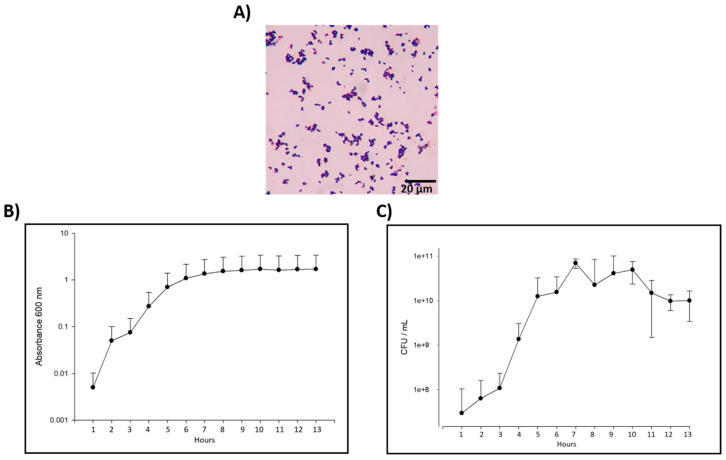
Morphology analysis, growth kinetics, and viability of *E. faecalis*. (**A**) Morphology analysis through Gram staining of *E. faecalis* using optical microscopy. (**B**) Growth kinetics establishment of *E. faecalis* was performed, evaluating the absorbance at 600 nm. (**C**) Cell viability analysis of *E. faecalis* by colony-forming unit (CFU).

**Figure 3 microorganisms-11-02208-f003:**
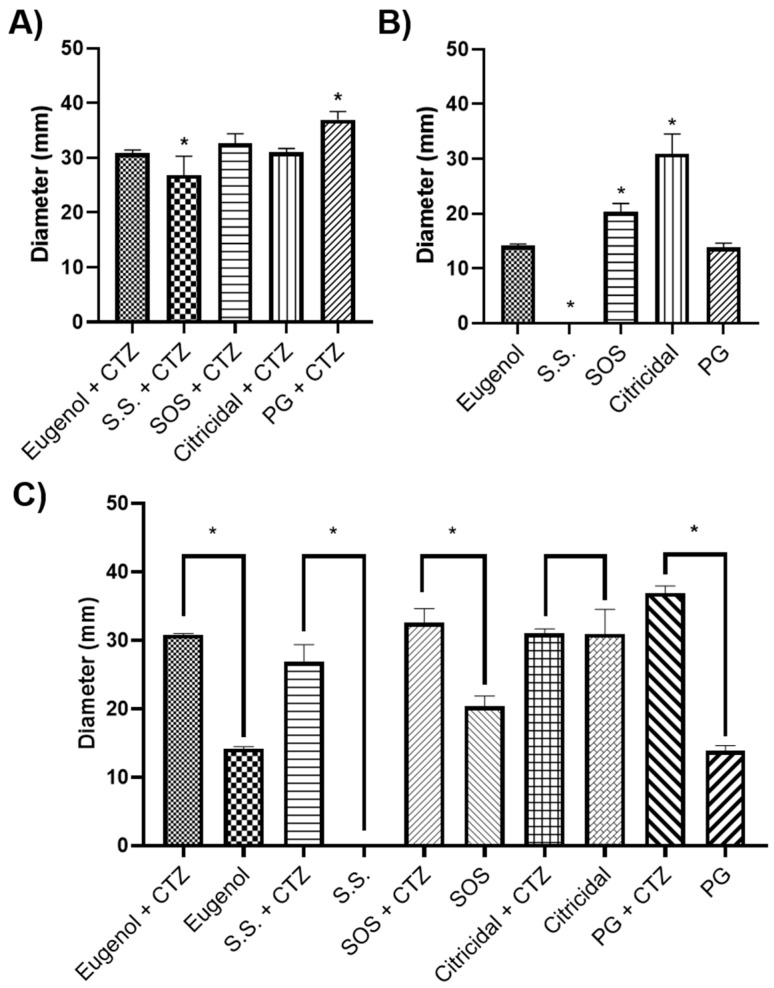
Evaluation of the effects of different vehicles mixed with CTZ paste or alone on *E. faecalis*. (**A**) Analysis of the inhibition of *E. faecalis* using the mixes of CTZ paste with the different vehicles. (**B**) Analysis of the inhibition of *E. faecalis* using the different vehicles. (**C**) Comparison analysis of *E. faecalis* inhibition using the mixes of CTZ paste with the different vehicles and the vehicles used alone. * Mean *p* < 0.05.

**Table 1 microorganisms-11-02208-t001:** Amount and vehicle concentration.

Vehicle	Amount	Concentration
Eugenol	200 µL	37.50%
Electrolyzed super oxidation solution	200	0.0015%
Grapefruit-seed extract	600 µL	46%
Propylene glycol	400 µL	99.50%
Saline solution (S.S.)	400 µL	0.90%

**Table 2 microorganisms-11-02208-t002:** Biochemical test characterization of *E. faecalis*.

Substrates	Result
Crystal violet	+
Micrococcus screen	+
Nitrate	−
Novobiocin	+
PNP-β-D-Glucuronide	−
Indoxyl phosphatase	−
Voges-Proskaure	−
Optochin	+
Phosphatase	+
40% Bils esculin	+
L-Pyrrolidonyl-β-naphtothylamide	+
Arginins	+
PNP-β-D-galactopyranoside	+
Urea	−
Mannitol	+
Lactose	+
Trehalose	+
Mannose	+
Sodium Chloride 6.5%	+
Sorbitol	+
Arabinose	−
Ribose	+
Inulin	-
Raffinose	−
Bacitracin	+
Pyruvate	+

**Table 3 microorganisms-11-02208-t003:** Antibiotics resistance test performed on *E. Faecalis*. * At the discretion of medical interpretation.

Antimicrobial Agent	MIC (µg/mL)	Interpretation
Amoxicillin clavulanic acid	≤4/2	*
Ampicillin sulbactam	≤8/4	*
Ampicillin	≤2	Sensitive
Ceftriaxone	>32	*
Ciprofloxacin	≤1	Sensitive
Clindamicina	>4	*
Daptomycin	2	Sensitive
Erythromycin	>4	Resistance
Erythromycin synergy	>1000	Resistance
Gentamicin synergy	≤500	Sensitive
Gentamicin	8	*
Levofloxacin	2	Sensitive
Linezolid	4	Intermediate
Moxifloxacin	≤0.5	*
Nitrofurantoin	≤32	*
Oxacillin	>2	*
Penicillin	2	Sensitive
Rifampicin	≤1	Sensitive
Tetracycline	>8	Resistance
Trimethoprim sulfamethoxazole	>2/38	*

## Data Availability

The data presented in this study are openly available in ResearchGate at DOI: 10.13140/RG.2.2.28208.12805.

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
