# Peer review of "Propylene Glycol Potentiates the Inhibitory Action of CTZ Paste on Antibiotic-Resistant Enterococcus faecalis Isolated from the Root Canal: An In Vitro Study"

_microorganisms, 2023, doi:10.3390/microorganisms11092208_

Round 1

Reviewer 1 Report

The manuscript would be of general interest to the researchers of this field, but the manuscript lacks some information and novelty that the author should consider and incorporate in the present form of the manuscript. Here are some basically concerns that need to be addressed in the present form of the manuscript.

Overall, the manuscript has a lot of punctuation and grammatical errors and needs to be corrected 

 At the introduction section, the authors should declare the importance of nature and natural products with some important facts and/or examples.

Overall, the manuscript has a lot of punctuation and grammatical errors and needs to be corrected 

Author Response

Dear reviewer we appreciated your observations and suggestions to improve this manuscript.

Overall, the manuscript has a lot of punctuation and grammatical errors and needs to be corrected 

RESPONSE: We appreciated your suggestion, punctuation and grammatical errors were corrected.

 At the introduction section, the authors should declare the importance of nature and natural products with some important facts and/or examples.

RESPONSE: We appreciated your suggestion; we attend this comment and the information was included.

Comments on the Quality of English Language Overall, the manuscript has a lot of punctuation and grammatical errors and needs to be corrected 

RESPONSE: We appreciated your suggestion, punctuation and grammatical errors were corrected.

Reviewer 2 Report

1.improve the introduction 

2.improve the discussion part vivdly 

3.what is the role of chitosan 

4.author should be add the prepared paste images

minor revision need

Author Response

Dear reviewer we appreciated your observations and suggestions to improve this manuscript.

1.improve the introduction 

RESPONSE: We appreciated your suggestion; we attend this comment an improve of the introduction was realized.

2.improve the discussion part vivdly 

RESPONSE: We appreciated your observation, an improve of the discussion was realized.

3.what is the role of chitosan 

RESPONSE: We appreciated your observation; all chitosan information was eliminated.

4.author should be add the prepared paste images

RESPONSE: We appreciated your suggestion; a figure with paste images was included.

Comments on the Quality of English Language.  minor revision need

RESPONSE: We appreciated your suggestion; an English revision of the manuscript was performed.

Reviewer 3 Report

The authors present a study investigating if changing the carrier vehicle for CTZ paste (chloramphenicol, tetracycline, zinc oxide) can improve the inhibition of Enterococcus faecalis in vitro. This is of interest because of E. faecalis' role in dental caries, one of the most prevalent diseases of the oral cavity. A new strain of E. faecalis was characterized using biochemical tests and MIC testing, and used to assess the inhibition zones produced by CTZ mixed with the vehicles eugenol, propylene glycol, chitosan, super-oxidised solution or grapefruit-seed extract (GSE), with 0.9% sterile saline used as the control vehicle. Propylene glycol + CTZ showed increased inhibition over the standard formulation, and GSE alone produced a large inhibition which is worthy of further investigation.

The Chitosan test should be repeated at a different concentration, as the authors stated the paste used was too thick for the active compounds to disperse. 

Some improvements to the manuscript are needed:

Table 1 - What are the units for the amount of Electrolysed super oxidation solution? And is the concentration really 0.00%?

Figure 2 - Make the scale of all graphs in Figure 2 the same (50mm). And please use a consistent pattern for each compound (or compound+CTZ) across all three panels as this will make the figure easier to interpret.

Careful editing by a fluent English speaker is needed for almost every sentence.

Careful editing by a fluent English speaker is needed for almost every sentence.

Pay careful attention to "by" and "for",  "because" and "since". eg. line 50 should be

'Antibiotic pastes have been proposed in endodontic therapy BECAUSE OF their antimicrobial capacity and low cost.' 

Also some sentences are very long or have too many clauses - it will be clearer if you use shorter sentences. eg. line 43 - 48. 

Check the tenses and flow of all your sentences eg. line 65 - 68. It would be better as:

'In the clinic several other types of paste are used to disinfect the root canal system and also to create a favorable environment for endodontic regeneration.  Triple Antibiotic Paste (TAP), which is composed by ciprofloxacin, metronidazole, and minocycline, is one example that is well studied. Studies have shown that the vehicles used with TAP can influence cytotoxicity, for example ....'

Author Response

Dear reviewer we appreciated your observations and suggestions to improve this manuscript.

The Chitosan test should be repeated at a different concentration, as the authors stated the paste used was too thick for the active compounds to disperse. 

RESPONSE: We appreciated your observation; all chitosan information was eliminated.

Some improvements to the manuscript are needed:

Table 1 - What are the units for the amount of Electrolysed super oxidation solution? And is the concentration really 0.00%?

RESPONSE: We appreciated your observation, the exact concentration.

Figure 2 - Make the scale of all graphs in Figure 2 the same (50mm). And please use a consistent pattern for each compound (or compound+CTZ) across all three panels as this will make the figure easier to interpret.

RESPONSE: We appreciated your observations; all modifications were realized.

Careful editing by a fluent English speaker is needed for almost every sentence.

 RESPONSE: We appreciated your suggestion; an English revision by a fluent English speaker was performed.

Comments on the Quality of English Language

Careful editing by a fluent English speaker is needed for almost every sentence.

Pay careful attention to "by" and "for", "because" and "since". eg. line 50 should be

'Antibiotic pastes have been proposed in endodontic therapy BECAUSE OF their antimicrobial capacity and low cost.' 

 Also some sentences are very long or have too many clauses - it will be clearer if you use shorter sentences. eg. line 43 - 48. 

 Check the tenses and flow of all your sentences eg. line 65 - 68. It would be better as:

'In the clinic several other types of paste are used to disinfect the root canal system and also to create a favorable environment for endodontic regeneration.  Triple Antibiotic Paste (TAP), which is composed by ciprofloxacin, metronidazole, and minocycline, is one example that is well studied. Studies have shown that the vehicles used with TAP can influence cytotoxicity, for example ....'

RESPONSE: We appreciated your suggestion; an English edition of the manuscript was performed.

Round 2

Reviewer 1 Report

The authors made corrections to the manuscript, in my opinion it can be accepted